# mTOR and STAT3 Pathway Hyper-Activation is Associated with Elevated Interleukin-6 Levels in Patients with Shwachman-Diamond Syndrome: Further Evidence of Lymphoid Lineage Impairment

**DOI:** 10.3390/cancers12030597

**Published:** 2020-03-05

**Authors:** Antonio Vella, Elisabetta D’Aversa, Martina Api, Giulia Breveglieri, Marisole Allegri, Alice Giacomazzi, Elena Marinelli Busilacchi, Benedetta Fabrizzi, Tiziana Cestari, Claudio Sorio, Gloria Bedini, Giovanna D’Amico, Vincenzo Bronte, Antonella Poloni, Antonio Benedetti, Chiara Bovo, Seth J. Corey, Monica Borgatti, Marco Cipolli, Valentino Bezzerri

**Affiliations:** 1Unit of Immunology, Azienda Ospedaliera Universitaria Integrata, 37134 Verona, Italy; antonio.vella@univr.it (A.V.); alice.giacomazzi@univr.it (A.G.); tiziana.cestari@univr.it (T.C.); vincenzo.bronte@univr.it (V.B.); 2Department of Life Sciences and Biotechnology, University of Ferrara, 44100 Ferrara, Italy; elisabetta.daversa@unife.it (E.D.); giulia.breveglieri@unife.it (G.B.); monica.borgatti@unife.it (M.B.); 3Cystic Fibrosis Center, Azienda Ospedaliero Universitaria Ospedali Riuniti, 60126 Ancona, Italy; martina.api@ospedaliriuniti.marche.it (M.A.); marisole.allegri@ospedaliriuniti.marche.it (M.A.); benedetta.fabrizzi@ospedaliriuniti.marche.it (B.F.); 4Hematology Clinic, Università Politecnica delle Marche -AOU Ospedali Riuniti, 60126 Ancona, Italy; e.busilacchi@staff.univpm.it (E.M.B.); Antonella.poloni@ospedaliriuniti.marche.it (A.P.); 5Department of Medicine, University of Verona, 37134 Verona, Italy; claudio.sorio@univr.it; 6Immunology and Cell Therapy Unit, Tettamanti Research Center, University of Milano-Bicocca, 20900 Monza, Italy; gloriabedini85@gmail.com (G.B.); giovanna.damico@asst-monza.it (G.D.); 7Department of Gastroenterology and Hepatology, Università Politecnica delle Marche, 60126 Ancona, Italy; antonio.benedetti@ospedaliriuniti.marche.it; 8Hospital Health Direction, Azienda Ospedaliera Universitaria Integrata, 37126 Verona, Italy; Direzione.sanitaria@aovr.veneto.it; 9Department of Pediatric Hematology/Oncology and Stem Cell Transplantation, Cleveland Clinic, Cleveland, OH 44195, USA; coreys2@ccf.org; 10Biotechnology Center, University of Ferrara, 44100 Ferrara, Italy; 11Cystic Fibrosis Center, Azienda Ospedaliera Universitaria Integrata, 37126 Verona, Italy; marco.cipolli@aovr.veneto.it

**Keywords:** STAT3, mTOR, Bone Marrow Failure Syndromes, lymphocytes

## Abstract

Shwachman–Diamond syndrome (SDS) is a rare inherited bone marrow failure syndrome, resulting in neutropenia and a risk of myeloid neoplasia. A mutation in a ribosome maturation factor accounts for almost all of the cases. Lymphoid involvement in SDS has not been well characterized. We recently reported that lymphocyte subpopulations are reduced in SDS patients. We have also shown that the mTOR-STAT3 pathway is hyper-activated in SDS myeloid cell populations. Here we show that mTOR-STAT3 signaling is markedly upregulated in the lymphoid compartment of SDS patients. Furthermore, our data reveal elevated IL-6 levels in cellular supernatants obtained from lymphoblasts, bone marrow mononuclear and mesenchymal stromal cells, and plasma samples obtained from a cohort of 10 patients. Of note, everolimus-mediated inhibition of mTOR signaling is associated with basal state of phosphorylated STAT3. Finally, inhibition of mTOR-STAT3 pathway activation leads to normalization of IL-6 expression in SDS cells. Altogether, our data strengthen the hypothesis that SDS affects both lymphoid and myeloid blood compartment and suggest everolimus as a potential therapeutic agent to reduce excessive mTOR-STAT3 activation in SDS.

## 1. Introduction

Shwachman–Diamond syndrome (SDS) is one of the most common inherited bone marrow failure syndromes (IBMFS), occurring in almost 1 out of 75,000 live births [1]. SDS results from biallelic mutations in the Shwachman–Bodian–Diamond syndrome gene (*SBDS*), which encode the SBDS protein. SBDS protein cooperates with its partner elongation factor-like GTPase 1 (EFL1) to catalyze the release of the ribosomal anti-association factor eIF6, facilitating the assembly of the functional 80S ribosome [2,3,4]. The IBMFS are also cancer predisposition syndromes, in particular myelodysplastic syndrome (MDS) and acute myeloid leukemia (AML). In the general population, MDS has an incidence ranging from 2–12 cases per 100,000 people, which increases as individuals age [5]. Patients with SDS demonstrate a risk of evolution to MDS of 8.1% and 36% at 10 and 30 years, respectively [6]. A recent genomic analysis of 1514 patients with MDS who underwent a stem cell transplant showed that 4% of the young adult patients had undiagnosed compound heterozygous mutations in *SBDS*, suggesting that SDS prevalence among MDS/AML patients may be underestimated [7]. AML derives from dysregulated proliferation and accumulation of immature myeloid progenitor cells into the bone marrow and peripheral blood, which finally leads to a severe impairment of the hematopoietic system. Acute leukemias rapidly disseminate after initial inception, escaping the anti-leukemic immunity process. Regulatory T cells play a key role in the maintenance of immune tolerance, which acts as a regulator of the tumor immunity [8]. CD4/CD8 double negative (DN) T cells have gained prominence among T regulatory cell subsets engaged in immunosurveillance. DN T cells are mature T cells representing almost 3–5% of the total peripheral T cell population [9]. Most human and murine T cells express and rearrange the α and β chains of the T cell receptor (TCR) and are recognized as TCRαβ T cells, whereas a small part of T cells do express the γ and δ chains, which are mostly DN T cells [9]. Interestingly, DN T cells showed anti-leukemic activity and synergy with conventional chemotherapies both in vitro and in patient-derived xenograft models of AML [10,11]. In a mouse model of AML, the leukemic cells promoted T cell tolerance with suppression of anti-tumor CD8^+^ T cells [12]. Failure of T cell-mediated anti-cancer immune response is associated with disease progression and poor outcome in MDS and AML.

IL-6/JAK/STAT3 signaling axis plays a key role in leukemogenesis [13]. The genes encoding the kinase protein JAK2 have indeed often been mutated in myeloproliferative disorders, leading to constitutive hyper-activation of its downstream effector, the transcription factor STAT3 [14]. Of note, STAT3 hyper-activation has been found in tumor-infiltrating leukocytes, in which STAT3 orchestrates the crosstalk between cancer and immune cells [15]. Furthermore, it has been reported that STAT3 inhibition in the myeloid compartment remarkably induces the anti-tumor capabilities of T cells and promotes their expansion in vivo [16,17]. STAT3 activation in immune cells is indeed associated with suppression of anti-tumor immunity. STAT3 excessive activation may be triggered by elevated levels of IL-6 present in the serum or released within the tumor microenvironment [14]. IL-6 may act in an autocrine or paracrine manner. IL-6 binds to its IL-6 receptor (IL-6R), localized onto the plasma membrane (membrane bound (mb)IL-6R), which is physically associated with the gp130 protein. This process is recognized as classical IL-6 signaling and leads to gp130 homodimerization, resulting in the activation of the IL-6 receptor complex [18]. In addition, IL-6 binds to the small extracellular secretory soluble IL-6 receptor (sIL-6R), which is generated by alternative splicing of IL-6R gene or by metalloproteinase-dependent cleavage of mbIL-6R. The sIL-6R mediates JAK-STAT3 activation in gp130 positive cells, which do not express mbIL-6R through a process termed IL-6 trans-signaling [19]. The IL-6 trans-signaling pathway has been reported in murine hematopoietic stem cells [20,21]. Elevated levels of IL-6 have been found in adult bone marrow niche of patients with AML [22]. Serum IL-6 levels were found to be significantly increased in pediatric patients with AML [23]. Moreover, increased IL-6 serum levels are associated with poor prognosis in several types of cancer, including AML [23]. Interestingly, *IL-6* gene expression can be regulated by STAT3 itself, resulting in a feedforward autocrine feedback loop [14].

STAT3 has been reported as a direct substrate for the mammalian target of rapamycin (mTOR), which induces STAT3 S727 phosphorylation [24,25]. In addition, the mTOR-inhibitor rapamycin inhibits STAT3 S727 phosphorylation [24]. Moreover, we have previously shown that mTOR can promote STAT3 phosphorylation both at residue tyrosine 705 and serine 727 in SDS leukocytes [26]. Previous studies have reported that relapse of AML is associated with the gain of additional mutations in the mTOR gene, often due to the cytotoxic chemotherapy received by patients [27,28]. Inhibition of the mTOR pathway using rapamycin or other analogue molecules, including everolimus (RAD001) as anti-leukemic agent, has shown potent anti-cancer capabilities both in vitro and in vivo [29,30].

To date, no pharmacological therapy has been developed for IBMFS-related MDS or AML, and allogeneic hematopoietic stem cell transplantation remains the unique option in these cases. Unfortunately, its efficacy is limited by the morbidity and mortality associated with graft-versus-host disease.

Here we show further analysis of the mTOR-STAT3 axis in an extended panel of lymphocytic populations including CD4^+^, CD8^+^, T cells, DN T cells, γδT cells, and Natural Killer cells (NK). Moreover, we have checked the expression of IL-6 in different cellular and clinical SDS models, including lymphoblastoid cell lines (LCL), bone marrow mononuclear hematopoietic progenitors (BM-MNC) and mesenchymal stromal cells (BM-MSC), and plasma obtained from an enlarged cohort of 31 patients with SDS (Table 1). Our data indicate that everolimus can restore a normal level of mTOR and STAT3 activation in primary SDS lymphocytes. Importantly, mTOR-STAT3 inhibition was paralleled by a downregulation of IL-6 expression in hematopoietic SDS cells. Our results suggest the existence of a mTOR-STAT3-IL-6 loop of activation in hematopoietic SDS cells, which may affect both myeloid and lymphoid compartment, thus contributing to malignant transformation over the time. Taken together, these data strengthen the hypothesis of the involvement of lymphoid lineage in SDS and suggest everolimus or new rapalogs as potential therapeutic agents in SDS patients.

## 2. Results

### 2.1. mTOR-STAT3 Pathway is Hyper-Activated Also in SDS Lymphocyte Subsets and Everolimus Can Reduce This Process In Vitro

The JAK/STAT3 pathway regulates T cell cytotoxic gene expression, proliferation, and survival. STAT3 inhibition in the myeloid compartment displays a remarkable induction of the T cell anti-tumor capabilities and promotes their expansion in vivo [16,17]. Thus, STAT3 activation in immune cells is associated with suppression of anti-tumor immunity. The protein kinase mTOR acts as an activator of the STAT3 pathway [24,25,31]. Accordingly, we recently reported that mTOR can activate STAT3 pathway in B cells, neutrophils and monocytes from SDS patients [26]. To assess the activation of mTOR-STAT3 pathway in SDS lymphoid compartment, we determined the phosphorylation of mTOR (S2448) and STAT3 (Y705 and S727) in CD4+ and CD8+ T cells, DN T cells, γδT cells, and NK cells isolated from peripheral blood from seven patients with SDS. In SDS patients, all cell subsets except NK displayed significantly elevated levels of phospho-mTOR (Figure 1) and -STAT3 (Figure 2) compared to age-matched healthy donors. Then, we tested the effect of the clinically-approved rapamycin analog, everolimus (RAD001) on mTOR, and STAT3 phosphorylation in SDS patient-derived T cell subsets. Results show that everolimus restores normal levels of phosphorylation of mTOR (Figure 1) and STAT3 (Figure 2), confirming the existence of an mTOR-STAT3 axis activation in the lymphoid compartment of SDS patients. To determine whether upregulation *STAT3* gene expression eventually exists in SDS cells along with hyper-phosphorylation, we measured and correlated *STAT3* transcript expression with protein levels in LCL and primary PBMC isolated from several SDS patients with different genotypes, compared to age-matched healthy controls. Our studies showed that *STAT3* expression is elevated in lymphocytes obtained from SDS patients compared to control subjects (Figure 3 and Figure 4, Appendix A).

### 2.2. IL-6 Expression Is Upregulated in SDS

Plasma levels of IL-6 are generally close to the detection limit (1 pg/mL) in healthy individuals but significantly increase during the inflammatory process and cancers [32]. IL-6 is a major activator of JAK-STAT3 signaling, and *IL*-*6* transcript expression is upregulated by STAT3 activation, generating an autocrine/paracrine loop of activation [14]. Since mTOR-STAT3 axis is upregulated in SDS patient-derived myeloid cells [26], we sought to find out whether hyper-activation of this pathway is associated with *IL-6* over-expression in non-myeloid SDS cell models. Both LCL and BM-MSC obtained from patients with SDS displayed significantly upregulated IL-6 release into cell culture supernatants compared with age-matched healthy controls (Figure 5a,b). In particular, IL-6 levels were significantly elevated in primary SDS BM-MSC, which released ~8 ng/mL in 2 × 10^5^ cells in our experimental conditions (Figure 5b). Plasma samples obtained from the peripheral blood collected from an expanded cohort of 21 patients with SDS showed significantly increased levels of IL-6 (mean 3.66 ± 4.58) compared to aged-matched controls (mean 1.19 ± 1.89), consistent with *in vitro* results (Figure 5c). Since SDS BM-MSC showed an impressive upregulation of *IL-6* expression, we sought to find out whether IL-6 were further concentrated in plasma derived from the bone marrow of patients. IL-6 levels in bone marrow plasma were even more elevated (mean 4.75 ± 3.82) than those found in peripheral blood (mean 3.04 ± 2.05) obtained in parallel, from the same patients (Figure 5d).

### 2.3. Patients with SDS ShowReduced Levels of Soluble IL-6 Receptor

IL-6 signaling cascade occurs by classical activation through mbIL-6R or trans-signaling via the soluble sIL-6R [19]. The latter mechanism allows cells that do not express mbIL-6R, but do express gp130, to be responsive to IL-6. Undifferentiated MSC lack the expression of IL-6 receptor although they normally express gp130 [33]. Thus, trans-signaling should be required to activate these cells upon IL-6 stimulation. Nevertheless, BM-MSC can constitutively release large quantities of IL-6 [34]. In this work, we quantified sIL-6R in plasma samples obtained from 21 patients with SDS. In healthy individuals, plasma levels of sIL-6R range between 50–70 ng/mL [35]. Of note, we found that the soluble receptor release is reduced in SDS patients (44.3 ± 15.4 pg/μL) compared to age-matched healthy donors (71.6 ± 14.7 pg/μL) (Figure 6a). To verify whether the bone marrow compartment, which showed increased levels of IL-6, exhibits also higher levels of sIL-6R, we compared sIL-6R expression in peripheral blood plasma with the expression found in bone marrow plasma obtained in parallel from the same patients. However, peripheral blood and bone marrow plasma showed comparable levels of sIL-6R expression (Figure 6b).

### 2.4. Elevated IL-6 Gene Expressionin Hematopoietic CellsIs Primarily Driven by mTOR-STAT3 Pathway in SDS

Fifty to eighty percent of patients with AML present a constitutive activation of the mTOR pathway, showing significantly shorter disease-free and overall survival rates compared with patients without constitutive activation [36,37]. In the last decade, the development of new rapamycin (sirolimus) analogs showing improved pharmacokinetic profile, such as the clinically approved rapalog everolimus, have given rise to great interest for anti-cancer therapy [29]. We previously reported that rapamycin-dependent mTOR inhibition leads to normal levels of phosphorylation of STAT3 in SDS cells [26]. Here we show that everolimus (350 nM) and STAT3 inhibitor stattic (7.5 μM) significantly reduce *IL-6* mRNA expression in LCL and in primary BM-MNC obtained from patients with SDS (Figure 7a,c). Decreased *IL-6* mRNA expression is paralleled by a reduction of IL-6 release in culture supernatants upon everolimus and stattic treatments both in LCL (-46.6% and -68%, respectively) and BM-MNC (-34.6% and -66%, respectively) (Figure 7b,d). In order to validate these data, we also knocked-down the expression of *mTOR* and *STAT3* in SDS cells using a short interference (si)RNA strategy. To this aim, we transfected two different siRNA sequences for each target gene and a negative control sequence (scrambled), which was previously validated [29], using siRNA-specific liposomes as a vector. We significantly knocked-down *mTOR* and *STAT3* gene expression in SDS LCL (Figure 8a,b). Consistently with pharmacological inhibition, both *mTOR* and *STAT3* gene silencing lead to a strong inhibition of *IL-6* expression in SDS cells (Figure 8c,d). In particular, knock-down of the *STAT3* gene resulted in a statistically significant reduction of *IL-6* expression in terms of mRNA (-85%) and protein release (-51%). However, no effect of everolimus nor stattic on IL-6 release was reported in BM-MSC obtained from four SDS patients (Figure 7f), and *IL-6* mRNA transcription was surprisingly increased (two-fold increase) upon stattic treatment in these cells (Figure 7e).

## 3. Discussion

Although it has been widely reported that SDS mainly involves the neutrophil lineage, a number of patients suffer from anemia, thrombocytopenia or pancytopenia (Table 1). Because the bone marrow is often hypocellular, lymphoid and stromal cells may contribute to reduced blood cell formation and myelodysplasia. We recently described a severe deficit of T cells, in particular of DN T subpopulation in SDS patients [38]. Here we show that T cell subpopulations isolated from SDS patients display also hyper-activation of mTOR-STAT3 pathway. In addition, *STAT3* transcript and protein expression are markedly increased in PBMC and LCL obtained from SDS patients, confirming the involvement of the STAT3 pathway in lymphoid lineages. Given the key role of STAT3 in reducing T regulatory cell accumulation [39], our results could partially explain the reduced number of DN T cells observed in SDS patients [38]. Furthermore, STAT3 activation in lymphocytes is associated with T cell impaired functions [17] and reduced anti-tumor activity [16,40]. Accordingly, impaired functions of T cells have been previously described in SDS [41]. Thus, STAT3 pathway upregulation could lead to harmful consequences both on myeloid differentiation in the bone marrow and on innate and adaptive immunosurveillance mechanisms in SDS.

IL-6 is a major activator of STAT3, and *IL-6* transcript expression is itself a target of STAT3 [14]. Thus, we measured *IL-6* expression both at mRNA and protein levels in several SDS cell types. We found that *IL-6* expression is elevated in LCL, primary BM-MNC and BM-MSC, compared to age-matched healthy controls. BM-MSC produce large amounts of IL-6 even in unstimulated and undifferentiated conditions [34]. Accordingly, BM-MSC obtained from SDS patients released huge amounts of IL-6 protein into the supernatants (~8 ng/mL, from 2 × 10^5^ cells), that is an amount comparable with the dose of IL-6 commonly used to stimulate in vitro these cells (10 ng/mL). Interestingly, that concentration of IL-6 has been also reported as the driving force leading to a further increase of mTOR-STAT3 activation in SDS leukocytes [26], suggesting the existence of a feedforward autocrine/paracrine feedback loop between STAT3 and IL-6 in SDS bone marrow. In order to verify this hypothesis in clinical samples, we measured peripheral blood plasma levels of IL-6 in a cohort of 21 patients with SDS. We found that IL-6 levels in peripheral blood plasma are elevated in SDS patients. Then, we measured IL-6 released into bone marrow plasma and we found even more elevated cytokine levels. Since BM-MNC do release very low amounts of IL-6 compared to BM-MSC, we speculate that IL-6 accumulation into the bone marrow environment is mainly due to the contribution from the stromal compartment. The MSC compartment is involved in AML development, contributing to disease initiation both in animal models and in patients [42,43,44]. For instance, studies on BM-MSC obtained from patients with MDS or AML reported altered expression of several cytokines and other soluble pro-inflammatory mediators [45]. Thus, increased interleukin-6 levels in bone marrow of patients with SDS, together with mTOR-STAT3 axis hyper-activation in hematopoietic cells, may clarify the reason why these subjects are prone to develop AML. We previously reported that rapamycin (sirolimus) treatment reduced the phosphorylation of both mTOR (S2448) and STAT3 (Y705 and S727) in cell lines as well as in primary neutrophils, monocytes and B cells isolated from patients with SDS. Here, we show that the clinically approved rapalog, everolimus, can restore normal level of activation of both mTOR and STAT3 in SDS lymphocyte populations. Importantly, in this study we show that everolimus and a commercially available STAT3 chemical inhibitor, namely stattic, can significantly inhibit IL-6 release in SDS LCL and BM-MNC. In the light of these findings, novel clinically approved inhibitors of mTOR and STAT3 might be helpful in reducing pro-leukemic pathways in SDS hematopoietic cells. STAT3 inhibitors are being evaluated as chemotherapeutic agents in leukemias, due to their strong pro-apoptotic activity [13]. Although we used 7.5 μM stattic, which was a previously reported dose that can inhibit STAT3 without affecting lymphoid cell viability in vitro [46], we observed a stattic-dependent induction of late apoptosis in LCL (Appendix A). We cannot exclude the fact that the effect observed in this case may partially be due to pro-apoptotic activity. However, since everolimus reduced both STAT3 activation and *IL-6* expression in SDS LCL and BM-MNC without inducing pro-apoptotic processes (Appendix A), we can assume that mTOR-STAT3 pathway inhibition might be useful in reducing the excessive cytokine release. Both everolimus and stattic did not reduce the huge release of IL-6 from undifferentiated SDS BM-MSC, which remains one of the main sources of IL-6 within the bone marrow compartment. This finding suggests other regulatory pathways of *IL-6* gene expression exist in BM-MSC.

IL-6 can trigger JAK-STAT3 signaling activation through the direct binding to mbIL-6R or via the soluble sIL-6R (IL-6 trans-signaling), as result of alternative splicing or protease cleavage of mbIL-6R [19]. In order to evaluate whether the mTOR-STAT3-IL6 loop is generated by increased IL-6 trans-signaling in SDS, we measured sIL-6R levels in plasma samples obtained from 21 patients. However, results indicate that sIL-6R expression is reduced in SDS patients compared to age-matched healthy donors, thus suggesting that this loop is mainly generated via the classical IL-6 signaling in hematopoietic cells. Interestingly, ex vivo undifferentiated BM-MSC lack the expression of mbIL-6R, even if they normally do express the gp130 protein [33]. Besides, only a few cell types do express mbIL-6R and therefore are able to respond to IL-6 generating the IL-6-STAT3-IL6 loop of activation. Among these cells, major players are represented by macrophages, neutrophils, T-cells, and hepatocytes. On the contrary, gp130 is generally ubiquitously expressed [47,48]. It has been suggested that the lack of mbIL-6R is a regulatory mechanism of BM-MSC that inhibits the IL-6-dependent chondrogenic/osteogenic differentiation, thus maintaining the stemness [33]. During osteogenic differentiation, the expression of IL-6R indeed increases in BM-MSC, allowing the autocrine/paracrine activation of the IL-6-STAT3 signaling pathway [49]. Thus, reduced sIL-6R levels in plasma suggest that the impaired ossification reported in SDS [50] might be partially due to decreased IL-6 trans-signaling in BM-MSC. The lack of IL-6 trans-signaling might therefore also partially explain why everolimus and stattic cannot reduce *IL-6* expression in BM-MSC. Our results indicate that BM-MSC are unable to activate the autocrine/paracrine feedforward loop of mTOR-STAT3-IL6.

## 4. Materials and Methods

### 4.1. Human Subjects

Human samples were obtained and analyzed in accordance with the declaration of Helsinki, after written consent. All protocols were approved by the Ethics Committee of San Gerardo Hospital (Monza, Italy), approval No. 504 4/9/2012, Azienda Ospedaliera Universitaria Integrata (Verona, Italy), approval No. 658 CESC, and Azienda Ospedaliero Universitaria Ospedali Riuniti (Ancona, Italy), approval No. CERM 2018-82.

### 4.2. Plasma Isolation

Bone marrow and peripheral blood specimens were collected into EDTA-containing tubes from patients. Specimens were collected during a bone marrow harvest from healthy donors serving as donors for a related HLA-matched transplant, as permitted by the local hospital ethics committee and after informed consent was obtained. Peripheral blood and bone marrow plasma samples were prepared by centrifugation for 10 min at 600× *g* at 4 °C. To obtain platelet-poor plasma fractions, another centrifugation was performed for 15 min at 1800× *g*. Plasma specimens were then stored at −80 °C until analysis.

### 4.3. Cell Cultures

LCL, PBMC, BM-MNC, and BM-MSC were obtained from peripheral blood or bone marrow samples from31 patients with Shwachman–Diamond Syndrome. All patients enrolled in this study were diagnosed as affected by SDS on the basis of clinical and genetics criteria, and they were excluded if MDS/AML were present (Table 1). None of the patients underwent granulocyte colony-stimulating factor (G-CSF, filgrastim) nor steroids therapies. PBMC and BM-MNC were separated by stratification on Ficoll-Paque PLUS (density 1.077 g/mL, GE Healthcare, Waukesha, WI, USA) gradient and washed twice with PBS (Euroclone, Milan, Italy). Cells were seeded in 6-wells cell culture plates at 1 × 10^6^ BM-MNC/well in 1 mL RPMI-1640 Medium (Gibco, Waltham, MA, USA) supplemented with 10% FBS (Sigma-Aldrich, St. Louis, MO) and incubated in the presence or absence of 350 nM everolimus (Sigma-Aldrich) or 7.5 μM stattic (Selleckchem, Houston, TX, USA) at 37 °C in humidified atmosphere, with 5% CO_2_ for 24 h. Treated cell pellets were collected by centrifugation and the supernatant isolated and stored at −80 °C. BM-MSC were isolated after seeding 1.6 × 10^5^ BM-MNC cells/cm^2^ from fresh bone marrow, in Dulbecco’s Modified Eagle Medium low glucose (Gibco) supplemented with 10% of FBS, 1% L-glutamine and 1% penicillin/streptomycin. BM-MNC were incubated at 37 °C in humidified atmosphere, with 5% CO_2_ for 24 h. Subsequently, non-adherent cells were removed by washing with PBS and culture medium was finally replaced.BM-MSC were seeded at 4 × 10^3^ cell/cm^2^ and incubated at 37 °C in humidified atmosphere, with 5% CO_2_ since they reached 70–80% confluence. 1 × 10^5^ MSC were seeded in a 6-well plate and incubated for 48 h at 37 °C in humidified atmosphere, with 5% CO_2_. Eventually, treated cells were collected and the supernatant isolated and stored at −80 °C. To obtain LCL, B cells were seeded at a density of 3 × 10^6^ cells in 12-well cell culture plates, incubated with 3 mL RPMI-1640 supplemented with 10% FBS, and infected for 18 h with Epstein–Barr virus (EBV) released from marmoset blood leukocytes B95.8 virus-producer cell lines as previously described [26]. 1 × 10^6^ LCL were seeded in 6 well-cell culture plate in RPMI-1640 supplemented with 10% FBS and incubated for 48 h at 37 °C in humidified atmosphere, with 5% CO_2_. Eventually,1 × 10^6^ LCL were seeded in 6 well-cell culture plate in RPMI-1640 supplemented with 10% FBS, in the presence or absence of 350 nM everolimus or 7.5 μM stattic at 37 °C in humidified atmosphere, with 5% CO_2_ for 24 h. Treated cell pellets were collected by centrifugation and the supernatant isolated and stored at −80 °C. LCL, BM-MNC and MSC cultured in medium containing DMSO (Sigma, dilution 1:10,000) were used as negative control.

### 4.4. Flow Cytometry

Plasma was separated from peripheral blood samples derived from SDS patients or healthy subjects by centrifugation at 1500× *g* for 10 min. Red blood cells were lysed in 40 mL of solution containing 0.89% (*w/v*) NH_4_Cl, 0.10% (*w*/*v*) KHCO_3_ and 200 μM EDTA. Leukocytes were cultured in 6-well plates containing RPMI-1640 supplemented with 10% freshly prepared, heat-inactivated human plasma, in the presence or absence (DMSO) of 350 nM everolimus for 1h. Fifty to one-hundred-100 microliter aliquots of blood cells were incubated for 15 min at room temperature with combinations of the following fluorochrome-conjugated monoclonal antibodies: CD3-APC750, CD4-PC7, CD8-ECD, CD16-Pacific Blue, CD19-Chrome-Orange, CD45-APC700, and CD56-PC5 (Beckman-Coulter). Cells were gently centrifuged (600× *g*) for 10 min, washed with ice-cold PBS, fixed, and permeabilized with Intracellular Fixation and Permeabilization Buffer Set (eBioscience, San Diego, CA, USA), following the manufacturer’s recommendations. Subsequently, cells were washed with flow buffer and stained with anti-pS727-STAT3-PE, anti-Y705-STAT3-PE, anti-p-S2448-mTOR-PE or isotype control-PE conjugated antibodies for 30 min. Data acquisition was performed by a 10 color, three laser Navios flow cytometer (Beckman Coulter, Indianapolis, IN, USA). Analysis of the acquired data was performed by the “Navios” or Kaluza software, version 1.3 (Beckman Coulter, Indianapolis, IN, USA). In order to define different subsets of lymphocytes, gating strategy and data filtering were established as follows. Cell debris were excluded using a dot plot for morphological parameters (FS, SS). Lymphocytes were gated into the side scatter (SS low) and CD45 positive area. Within the lymphocyte compartment, CD3^+^ events were further gated into CD4^+^ and CD8^+^ T cells. NK cells were instead identified as CD3/CD19 double-negative events (CD56^+^ and/or CD16^+^). Double-negative T cells were gated by plotting T cells (CD3^+^) in CD4 versus CD8 dot plots. Since it has been reported that TCR γδ^+^T cells are recognizable by the bright expression on their membrane of the CD3 molecule [51], CD3 bright DN T cells (DN γδ) were gated by plotting CD3 bright positive area into CD4 versus CD8 dot plots. Flow cytometry was performed on at least 25,000 events. The total white blood cell count (WBC) was determined by Hematology Analyzer XN- 9000 (Sysmex Europe GmbH, Norderstedt, Germany).

### 4.5. qRT-PCR

Total RNA from LCL and MNC was isolated using High Pure RNA Isolation Kit (Roche, Mannheim, Germany) following the manufacturer’s instructions. RNA was eluted in 50 µL of RNAse free water. Final RNA concentration was determined using NanoDrop 2000 spectrophotometer (Thermo Fisher Scientific, Foster City, CA, USA) and then stored at −80 °C until use. A total amount of 500 ng of RNA was reverse transcribed to cDNA using a High Capacity cDNA Reverse Transcription Kit with random primers (Applied Biosystems by Thermo Fisher Scientific) for 120 min at 37 °C and 5 min at 85 °C in a final reaction volume of 20 μL. For real-time qPCR analysis, 5 Μl of cDNA were used for each reaction to quantify the relative gene expression. The cDNA was then amplified for 45 PCR cycles using Platinum SYBR Green qPCR Super Mix-UDG (Invitrogen by Thermo Fisher Scientific) in a 20 μL reaction using the Rotor-Gene 6000 cycler (Qiagen, Hilden, Germany). In order to perform the PCR reaction, QuantiTect Primer Assays (Qiagen) for IL-6 (Hs_IL6_1_SG, NM_000600), STAT3 (Hs_STAT3_1_SG, NM_003150), mTOR (Hs_MTOR_1_SG, NM_004958), and GAPDH (HS_GAPDH_1_SG, NM_001256799) were purchased. Each target gene expression was normalized with GAPDH gene expression and relative quantification was calculated by the ΔCt method.

### 4.6. IL-6 and sIL-6R Detection

Bio-Plex Pro Human IL-6 Assay (Bio-Rad Laboratories, Philadelphia, PA, USA) was used to measure IL-6 protein released into cell supernatants and plasma samples (sample volume 50 μL). The assays were performed using the Bio-Plex Suspension Array System, with the Bio-Rad 96-well plate reader. Data were analyzed by the Bio-Plex Manager software (Bio-Rad Laboratories). The in-vitro quantitative determination of soluble IL-6 Receptor (sIL-6R) in plasma was performed by using the Human IL-6R/CD126 ELISA Kit (Origene, Rockville, MD, USA), according to the manufacturer’s instructions. Plasma samples were assayed after 1:200 dilution.

### 4.7. Western Blot

A total of 30 μg of cell extract was denatured for 5 min at 95 °C in 4× Laemmli Sample Buffer, (277.8 mM Tris-HCl, pH 6.8, 44.4% glycerol, 4.4% LDS, 0.02% bromophenol blue) (Bio-Rad Laboratoires) supplemented with 355 mM 2-mercaptoethanol. The samples were loaded on 7.5% polyacrylamide SDS-PAGE gel in Tris-glycine Buffer (25 mM Tris, 192 mM glycine, and 0.1% SDS) using tag protein ladder (Spectra Multicolor Broad Range Protein Ladder, Thermo Scientific, Waltham, MA, USA) to determine molecular weight. The electrotransfer was performed into nitrocellulose membrane (iBlot Gel Transfer Stacks Nitrocellulose, Thermo Fisher) at 20V using iBlot Dry Blotting System (Invitrogen, Waltham, MA, USA) for 10 min. The membranes were blocked in BSA 5% for 90 min at room temperature and washed with TBS (5 mM Tris-HCl pH 7.6, 150 mM NaCl) supplemented with 0.05% tween-20 (TBS/T) for 15 min. Subsequently, membranes were probed with primary anti-human STAT3 rabbit polyclonal antibody (SAB2104912 Sigma-Aldrich, Missouri, USA, dilution 1:500) in primary antibody dilution buffer (TBS/T with 5% BSA) and incubated overnight at 4 °C. After washes, membranes were incubated with mouse anti-rabbit IgG- Horseradish Peroxidase-Coupled secondary antibody (Sigma-Aldrich, dilution 1:2000) diluted TBS/T for 90 min at room temperature. Immunocomplexes were visualized using chemiluminescence (ECL Plus Western Blotting Substrate, Pierce, Thermo Scientific), incubating ECL for 5 min at room temperature. Band intensity was calculated by scanning video densitometry using the Chemi Doc imaging system (UVP, LCC, Upland, CA, USA). Blots were re-probed with monoclonal β-Actin-Peroxidase clone AC-15 antibody (Sigma-Aldrich, dilution1:5000) in TBS/T for 90 min.

### 4.8. Gene Silencing

Both *mTOR* and *STAT3* genes were knocked-down by siRNA. The lipid-based agent for reverse transfection, siPORT NeoFX (Thermo Fisher), was used according to the manufacturer’s instructions. LCL derived from healthy donors and SDS patients were transiently transfected with two different specific siRNA sequences designed against mTOR and STAT3 genes or with a scrambled sequence as control. The siRNA molecules were complexed with cationic liposomes, siPORT Neo-FX (Thermo Fisher). Briefly, siPORT Neo-FX (1 μL/well) was complexed with siRNA or Scrambled oligos (40 nM each) in 250 µL RPMI-1640 medium supplemented with 0.5% FBS. 1 × 10^5^ LCL were incubated in 24-wells plates for 48 h at 37 °C. Knock-down of *mTOR* and *STAT3* gene expression was determined by Real Time qRT-PCR.

### 4.9. Statistical Analysis

Normal distribution was tested in each experiment using the Shapiro–Wilk test. Based on that evaluation, independent group determination was tested using Mann–Whitney test, while Student’s *t*-test was used in case of paired data. A *p*-value < 0.05 was considered statistically significant. The statistical software SigmaPlot (Systat Software Inc., San Jose, CA, USA) was used.

## 5. Conclusions

STAT3 acts as a double-edged sword in SDS cells, as this pathways controls myeloid progenitor growth and proliferation [52] and promotes leukemogenesis. Extending our findings on mTOR-STAT3 signaling dysregulation in myeloid lineage [26], we now show that constitutive activation of mTOR-STAT3 axis occurs in the lymphoid compartment of SDS patients. An autocrine or paracrine feedforward loop of STAT3-IL6 exists in hematopoietic SDS cells. Since the loss of *SBDS* expression in healthy donor-derived cells is sufficient to reproduce the hyper-activation of mTOR-STAT3 signaling [26], we assume that alteration of the mTOR-STAT3-IL6 axis could be used by SDS cells as a survival mechanism that induces cell proliferation and myeloid growth in bone marrow, thus trying to escape from incoming neutropenia/aplasia processes.

## Figures and Tables

**Figure 1 cancers-12-00597-f001:**
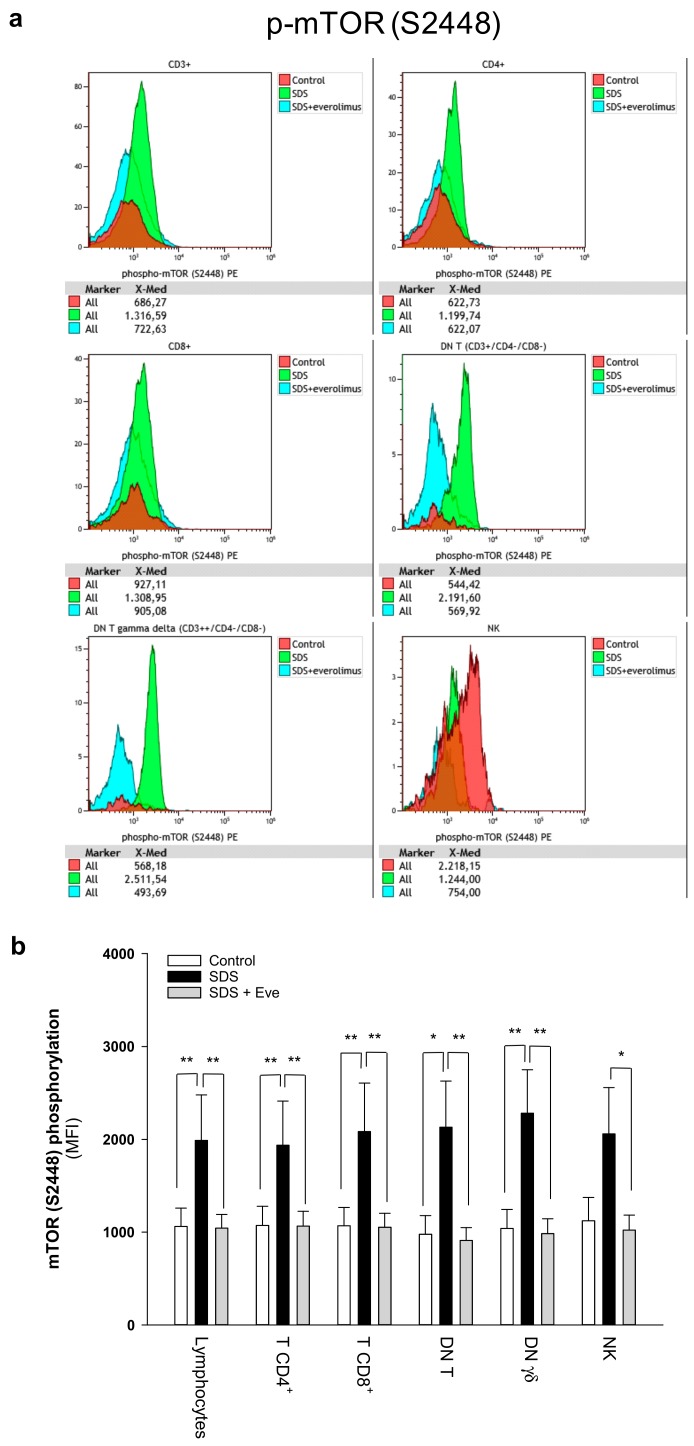
Phospho-flow analysis of mTOR S2448 in a panel of primary lymphocyte subsets. (**a**) Representative experiments conducted on peripheral blood samples obtained from patients with SDS. Red histograms represent age-matched healthy control cells; green histograms represent SDS lymphocytes; blue histograms represent SDS lymphocytes upon everolimus (350 nM) treatment. (**b**) Data are mean ± SEM of seven experiments conducted on SDS lymphocytes obtained from seven SDS patients (UPN37, UPN58, UPN69, UPN73, UPN87, UPN106, UPN108). Statistics: normal distribution was tested by the Shapiro–Wilk test. Subsequently, the Mann–Whitney Rank Sum Test was calculated. * *p* < 0.05; ** *p* < 0.01.

**Figure 2 cancers-12-00597-f002:**
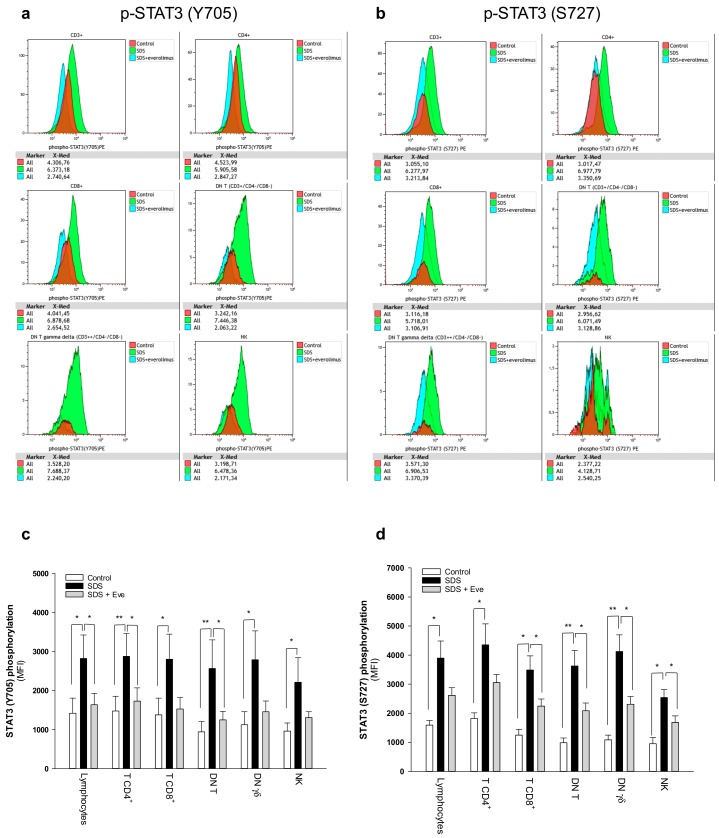
Phospho-flow analysis of STAT3 in a panel of primary lymphocyte subsets. (**a**) Representative analysis of phospho-STAT3 Y705 in lymphocyte subsets. (**b**) Representative analysis of phospho-STAT3 S727 in lymphocyte subsets. Red histograms represent age-matched healthy control cells; green histograms represent SDS patient-derived lymphocytes; blue histograms represent SDS lymphocytes upon everolimus (350nM) treatment. Data are mean ± SEM of seven experiments conducted on SDS patient-derived lymphocytes obtained from seven SDS patients (UPN37, UPN58, UPN69, UPN73, UPN87, UPN106, UPN108). Statistical Student’s *t* test for paired data has been calculated. * *p* < 0.05; ** *p* < 0.01. (**c**) STAT3 (Y705) Median Fluorescence Intensity as measured by phospho-flow assays. (**d**) STAT3 (S727) Median Fluorescence Intensity as measured by phospho-flow assays. Data are mean ± SEM of seven experiments conducted on SDS lymphocytes obtained from seven SDS patients (UPN37, UPN58, UPN69, UPN73, UPN87, UPN106, UPN108). Statistics: Normal distribution was tested by the Shapiro–Wilk test. Subsequently, the Mann–Whitney Rank Sum Test was calculated. * *p* < 0.05; ** *p* < 0.01.

**Figure 3 cancers-12-00597-f003:**
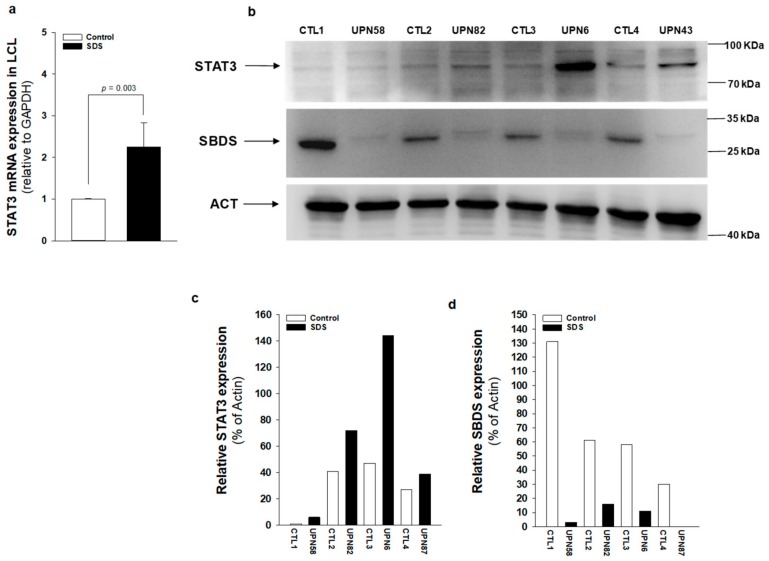
*STAT3* transcript and protein expression is upregulated in SDS patient-derived LCL. (**a**) *STAT3* mRNA expression in LCL isolated from UPN6, UPN43, UPN58, UPN82 (black bar), and from age-matched controls (white bar), measured by qRT-PCR. Data are mean ± SEM of four experiments performed in duplicate. (**b**) STAT3 protein level was measured in LCL (UPN6, UPN43, UPN58, UPN82) by Western blot analysis. (**c**,**d**) Densitometric analysis of Western blots showed in panel (**b**). Statistics: Normal distribution was tested by the Shapiro–Wilk test. Subsequently, the Mann–Whitney Rank Sum Test was calculated and reported within histograms.

**Figure 4 cancers-12-00597-f004:**
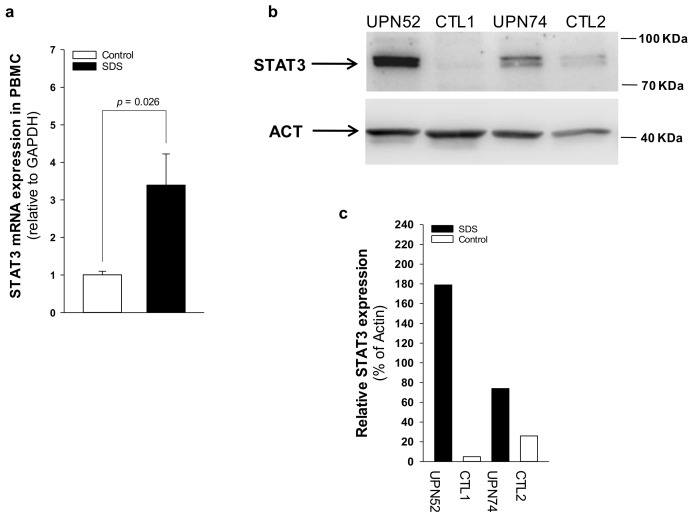
*STAT3* transcript and protein expression is upregulated in SDS patient-derived primary PBMC. (**a**) *STAT3* mRNA expression in primary PBMC isolated from UPN26, UPN69, UPN73, UPN87, UPN94, UPN106, and UPN108 (black bar), and from age-matched controls (white bar), measured by qRT-PCR. Data are mean ± SEM of seven experiments. (**b**) STAT3 protein level was measured in PBMC (UPN52 and UPN74) by Western blot analysis. (**c**) Densitometric analysis of Western blots showed in panel **b**. Statistics: Normal distribution was tested by the Shapiro–Wilk test. Subsequently, the Mann-Whitney Rank Sum Test was calculated and reported within histograms.

**Figure 5 cancers-12-00597-f005:**
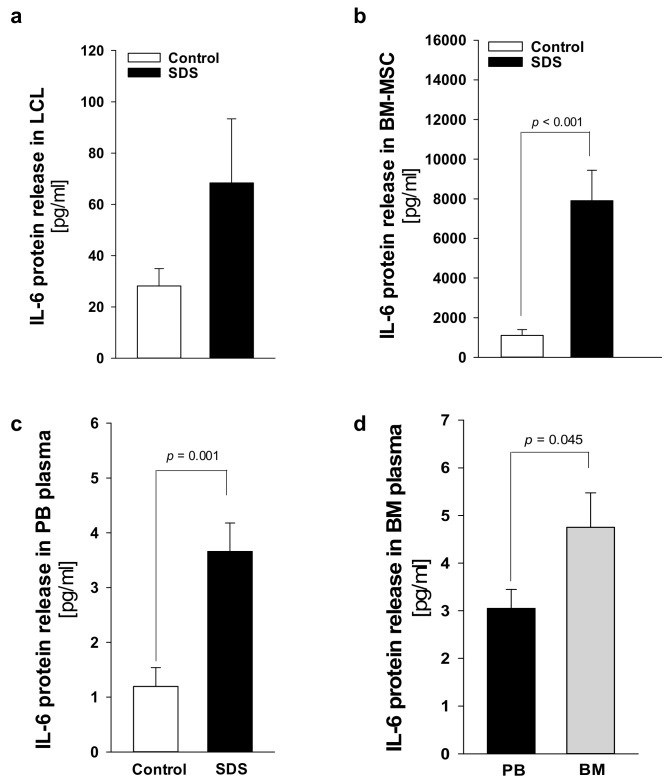
IL-6 release is elevated in SDS specimens compared to age-matched donor controls. (**a**) Measurement of IL-6 released in supernatants collected from 1 × 10^6^ LCL after 48h. Data are mean ± SEM of 10 experiments conducted on LCL obtained from UPN24, UPN26, UPN58, UPN68, UPN75, and UPN106. (**b**) IL-6 released in supernatants collected from 2 × 10^5^ primary BM-MSC after 48 h. (**c**) IL-6 concentration in peripheral blood (PB) plasma samples obtained from 21 patients with SDS (UPN1, 13, 26, 37, 47, 52, 56, 57, 58, 63, 65, 66, 69, 72, 73, 74, 87, 94, 104, 106, 108) compared with age-matched plasma controls. (**d**) IL-6 concentration in bone marrow (BM) plasma samples obtained from eight patients with SDS (UPN 47, 56, 65, 74, 87, 94, 106, 108) compared to PB plasma samples obtained from the same patients. Normal distribution was tested by the Shapiro–Wilk test. The Mann–Whitney Rank Sum Test was calculated and reported within histograms.

**Figure 6 cancers-12-00597-f006:**
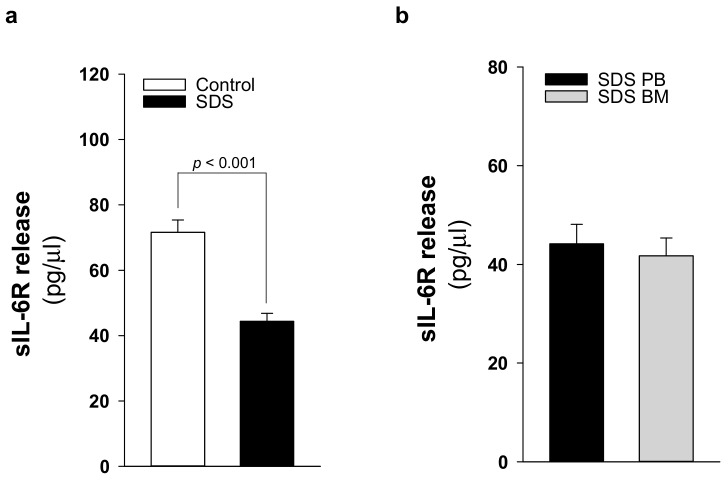
sIL6-R release is reduced in patients with SDS. (**a**) sIL-6R protein release was quantified by ELISA in PB plasma samples obtained from 21 patients with SDS (UPN 1, 13, 26, 37, 47, 52, 56, 57, 58, 63, 65, 66, 69, 72, 73, 74, 87, 94, 104, 106, 108) compared with age-matched plasma controls. (**b**) sIL-6R concentration in BM plasma samples obtained from eight patients with SDS (UPN 47, 56, 65, 74, 87, 94, 106, 108) compared with PB plasma samples obtained from the same patients. Normal distribution was tested by the Shapiro–Wilk test. The Mann–Whitney Rank Sum Test was calculated and reported within histograms.

**Figure 7 cancers-12-00597-f007:**
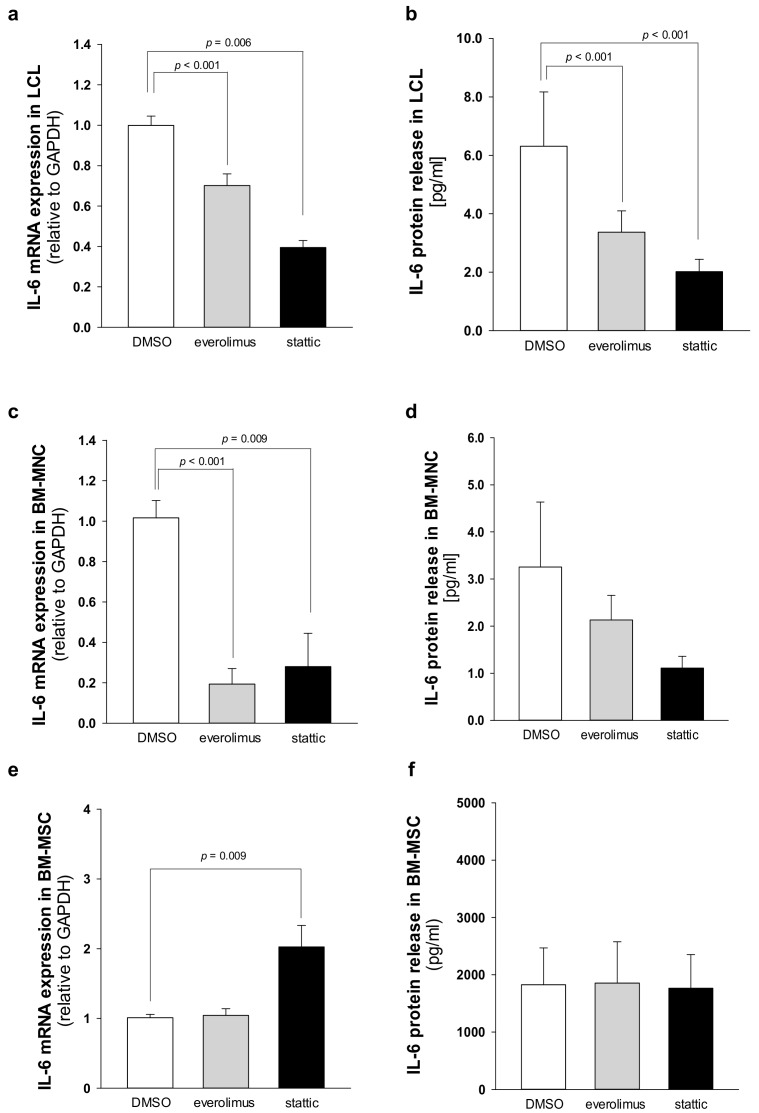
Everolimus and stattic inhibit *IL-6* expression in SDS patient-derived hematopoietic cells. (**a**) *IL-6* transcript expression in LCL incubated in the absence (DMSO) or in the presence of 350 nM everolimus or 7.5 μM stattic for 24 h was quantified by qRT-PCR. Data are mean ± SEM of six experiments performed in duplicate from three affected individuals (UPN58, UPN75, and UPN106). (**b**) IL-6 release in supernatants collected from LCL incubated in the absence (DMSO) or in the presence of 350 nM everolimus or 7.5 μM stattic for 24 h, as measured by Bio-plex assay. Data are mean ± SEM of six experiments conducted as reported in panel a. (**c**) *IL-6* mRNA expression in primary BM-MNC incubated in the absence (DMSO) or in the presence of 350 nM everolimus or 7.5 μM stattic for 24 h was quantified by qRT-PCR. Data are mean ± SEM of four experiments performed in duplicate from four affected individuals (UPN74, UPN80, UPN94, and UPN106). (**d**) IL-6 release in supernatants collected from BM-MNC as measured by Bio-plex assay. Data are mean ± SEM of four experiments conducted as reported in panel **c**. (**e**) IL-6 transcript expression in BM-MSC incubated in the absence (DMSO) or in the presence of 350 nM everolimus or 7.5 μM stattic for 24 h was quantified by qRT-PCR. Data are mean ± SEM of four experiments performed in duplicate from four affected individuals (UPN33, UPN35, UPN67 and UPN91). (**f**) IL-6 release in supernatants collected from BM-MSC (UPN33, UPN35, UPN67 and UPN91) as measured by Bio-plex assay. Data are mean ± SEM of four experiments conducted as reported in panel **e**. Statistics: Normal distribution was tested by the Shapiro–Wilk test, and the Student’s t test for paired data has been calculated and reported within histograms, accordingly.

**Figure 8 cancers-12-00597-f008:**
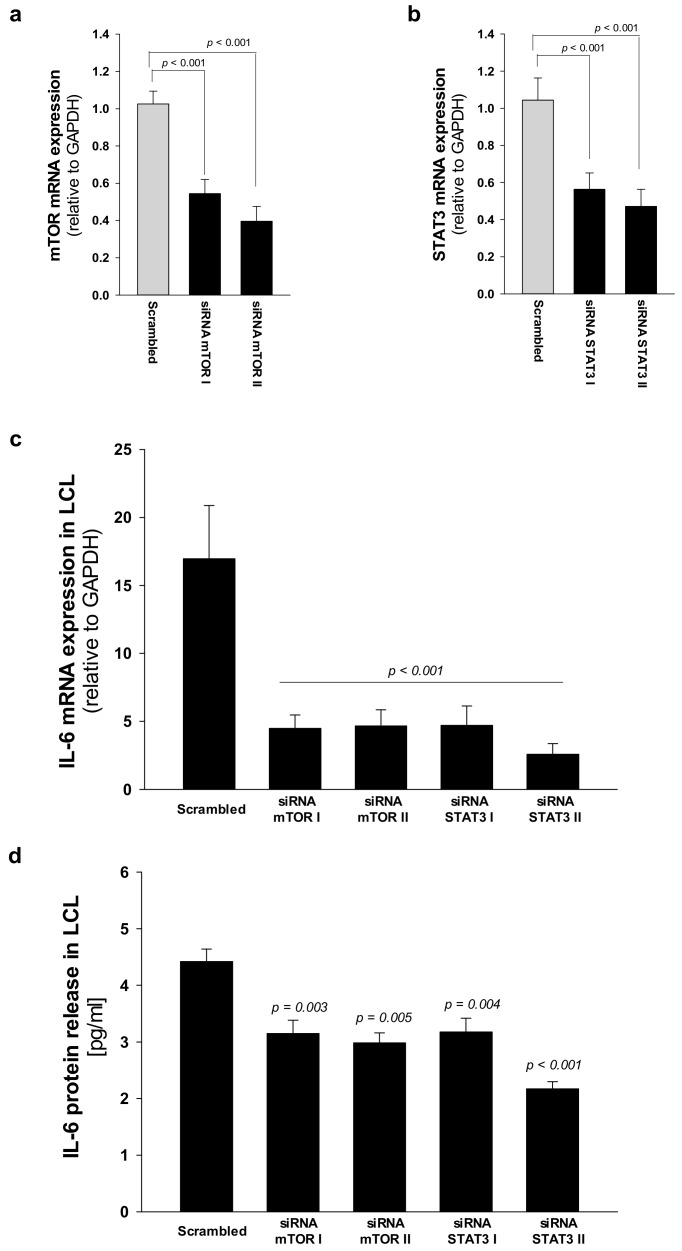
*STAT3* and *mTOR* gene silencing inhibit *IL-6* expression in LCL from SDS patients. (**a**) Reduced expression of *mTOR* mRNA after siRNA-mediated gene silencing. LCL obtained from UPN58, UPN75 and UPN106 were cultured with two different siRNA sequences against target genes, or scrambled sequence (negative control for 48 h). Data are mean ± SEM of four experiments performed in duplicate. (**b**) Reduced expression of STAT3 mRNA after siRNA-mediated gene silencing in LCL, as performed in panel a. Data are mean ± SEM of four experiments performed in duplicate. (**c**) *IL-6* mRNA expression in LCL treated as indicated in panels (**a**) and (**b**). (**d**) IL-6 release was measured by Bio-plex assay in supenatants obtained from LCL treated as indicated in **a**. Statistics: Normal distribution was tested by the Shapiro–Wilk test. Subsequently, the Mann–Whitney Rank Sum Test was calculated and reported within histograms.

**Table 1 cancers-12-00597-t001:** Clinical data and genetics of patients with SDS recruited in this study.

UPN	Gender	Age	Genotype	PMN (Cell/mm^3^)	Phenotype	Cytogenetics
1	M	27	258+2T>C/183–184TA>CT	1460	PI, FTT, recurrent infections, HbF > 2%, bone malformation, thrombocytopenia	46, XY, i(7)(q10)
6	M	27	258+2T>C/101A>T	3515	PI, FTT, recurrent infections, bone malformation, thrombocytopenia, cognitive impairment	46, XY, del(20)q
13	M	19	258+2T>C/183–184TA>CT+258+2T>C	1130	PI, FTT, recurrent infections, bone malformation, thrombocytopenia, anemia, cognitive impairment	46, XY, del(20)q
26	M	16	258+2T>C/183–184TA>CT	58	PI, FTT, bone malformation, thrombocytopenia	46, XY
33	F	8	258+2T>C/183–184TA>CT	1100	PI, FTT, bone malformation, thrombocytopenia	46, XX
35	F	15	258+2T>C/183–184TA>CT	970	PI, FTT, thrombocytopenia, anemia	46, XX
37	F	10	258+2T>C/183–184TA>CT	1280	PI, FTT, recurrent infections	46, XX, i(7)(q10)
43	M	22	258+2T>C/258+2T>C+533–549+403del	970	PI, FTT, bone malformation, thrombocytopenia, cognitive impairment	46, XY
47	M	12	258+2T>C/183–184TA>CT	550	PI, FTT, recurrent infections, bone malformation, thrombocytopenia	46, XY
52	M	10	258+2T>C/183–184TA>CT+258+2T>C	1070	PI, FTT, bone malformation	46, XY
56	F	15	258+2T>C/183–184TA>CT	1840	PI, FTT, recurrent infections, HbF > 2%, bone malformation, thrombocytopenia	46, XX
57	F	40	258+2T>C/G63C	500	PI, FTT, HbF > 2%, bone malformation, thrombocytopenia, cognitive impairment	46, XX
58	M	12	258+2T>C/183–184TA>CT	390	PI, FTT, HbF > 2%, bone malformation, thrombocytopenia, anemia	46, XY
63	M	15	258+2T>C/183–184TA>CT+258+2T>C	536	PI, FTT, recurrent infections, HbF > 2%, bone malformation, thrombocytopenia	46, XY
65	M	19	258+2T>C/258+2T>C	1390	PI, recurrent infections, bone malformation, thrombocytopenia, cognitive impairment	46, XY, del(20)q
66	M	23	258+2T>C/183–184TA>CT	1340	PI, bone malformation, thrombocytopenia, cognitive impairment	46, XY
67	M	8	258+2T>C/183–184TA>CT	500	PI, FTT, HbF > 2%, bone malformation	46, XY
68	M	22	258+2T>C/183–184TA>CT+258+2T>C	600	PI, FTT, recurrent infections, bone malformation, thrombocytopenia, cognitive impairment	46, XY
69	F	8	258+2T>C/183–184TA>CT	770	PI, FTT, HbF > 2%, anemia, cognitive impairment	46, XX
72	M	28	258+2T>C/183–184TA>CT	380	PI, FTT, recurrent infections, HbF > 2%, bone malformation, thrombocytopenia, anemia, cognitive impairment	46, XY
73	F	7	258+2T>C/183–184TA>CT	520	PI, FTT, HbF > 2%, thrombocytopenia	46, XX
74	M	9	258+2T>C/183–184TA>CT	1430	PI, FTT, HbF > 2%, cognitive impairment	46, XY
75	F	7	258+2T>C/183–184TA>CT	1000	PI, FTT, HbF > 2%, bone malformation, thrombocytopenia, cognitive impairment	46, XX
80	M	7	258+2T>C/183–184TA>CT	680	PI, FTT, recurrent infections, bone malformation, anemia, cognitive impairment	46, XY
82	M	16	258+2T>C/183–184TA>CT	300	PI, FTT, recurrent infections, bone malformation, thrombocytopenia, anemia, cognitive impairment	46, XY
87	M	18	258+2T>C/183–184TA>CT	880	PI, FTT, recurrent infections, bone malformation, cognitive impairment	46, XY
91	M	4	258+2T>C/183–184TA>CT+258+2T>C	1050	PI, FTT, bone malformation	46, XY
94	F	19	258+2T>C/352A>G	2420	PI, HbF > 2%, thrombocytopenia	46, XX
104	M	10	258+2T>C/183–184TA>CT	500	PI, FTT, recurrent infections, HbF > 2%, bone malformation, thrombocytopenia	46, XY
106	M	36	258+2T>C/183–184TA>CT	1210	PI, FTT, bone malformation, anemia	46, XY
108	M	17	258+2T>C/183–184TA>CT	970	PS, FTT, bone malformation, anemia	46, XY

UPN, unique patient number; PI, pancreas insufficiency; PS, pancreas sufficiency; FTT, failure to thrive; HbF, fetal hemoglobin.

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
