# Peer review of "mTOR and STAT3 Pathway Hyper-Activation is Associated with Elevated Interleukin-6 Levels in Patients with Shwachman-Diamond Syndrome: Further Evidence of Lymphoid Lineage Impairment"

_cancers, 2020, doi:10.3390/cancers12030597_

Round 1

Reviewer 1 Report

The MS has been much improved and I am satisfied with the answers to my comments.

Only one little bug:

Ref 26 (line 98)  is cited before ref 25 (line 131), please correct.

Author Response

We thank the Reviewer for his/her helpful revision of the manuscript, which now sounds remarkably improved.

Ref 26 (line 98)  is cited before ref 25 (line 131), please correct.

R1.1. Now ref 25 is cited in line 96 and subsequently, ref 26 is cited in line 98.

Reviewer 2 Report

The article studies lymphocyte populations in Shwachman-Diamond-Syndrome, an inherited bone marrow failure syndrome. The authors analyzed patient materials and patient cells for the activation of the previously noted mTOR-STAT3-IL-6 axis, which they show to be hyperactivated in the lymphoid compartment resulting in massive IL-6 production. Everolimus mediated inhibition of mTOR, knock-down of mTOR and knock-down of STAT3 alleviated this hyperactivation. The authors speculate that inhibition of mTOR by everolimus might be useful for the treatment of Shwachman-Diamond-Syndrome or similar malignancies.

The authors have responded to the points raised by this reviewer in a satisfactory way.

Author Response

We thank the Reviewer for his/her contribution. 

This manuscript is a resubmission of an earlier submission. The following is a list of the peer review reports and author responses from that submission.

Round 1

Reviewer 1 Report

This MS by Vella et al describes the hyperactivation of the mTOR and STAT3 pathways in multiple lymphoid cell lineages as well as in bone marrow-derived mesenchymal stromal cells of Shwachman-Diamond syndrome patients, leading to overproduction of IL-6 which in turn could be involved in determining the high risk of myeloid neoplasia characterizing these patients. Additionally, they show that the mTOR inhibitor everolimus as well as mTOR  or STAT3 silencing normalize STAT3 phosphorylation and IL-6 expression and secretion in several kinds of cells, suggesting that everolimus treatment could be of help in managing myeloid neoplasia development. This is an interesting paper and the results are mostly sound. However, there are several problems as detailed below.

In several parts of the manuscript the authors state that mTOR can promote phosphorylation of STAT3 on both Y-705 and S-727 but the references cited do not show this at all (in particular, reference 28 does not even mention STAT3!). Moreover, as far as this reviewer can remember, while there are several papers showing the involvement of mTOR in S-727 phosphorylation, none mentions Y-P, as one would expect. Please use more carefully the literature and moderate these statements. Fig 1: the authors state that mTOR phosphorylation is enhanced in all cell types analysed, while NK cells do not show any increase. Statistical analysis of data in Figs 1&2: why do the author use Mann-Whitney Rank Sum in Fig 1 and Student’s t test for paired data in Fig. 2? Please justify why not Mann-Whitney in both and unify the method. Moreover, the Kolmogorov-Smirnov test would perhaps be more suitable for both sets of experiments. Finally, characters in both figures are too small and the figures themselves are too low resolution. Both legends to figure 1 and 2 lack a correct description of the graphics c and d. Fig 3: why do the authors did not use the same samples for protein and mRNA detection of STAT3? Indeed, more samples analysed at the protein level would be desirable. Fig 4: please describe in the Methods how the bone marrow plasma was derived. Particularly since the authors emphasize the importance of the high IL6 secretion by BM-MSC making the point that the observed phenotype in patients could mainly be due to disrupted pathways in the stromal compartment of the bone marrow, it is essential to repeat the everolimus/STATTIC experiments of Fig. 5 and the siRNA experiments of Fig 6 in these cells as well. All figure’s labels are too small and almost unreadable, please modify. Finally, cells should be controlled for viability/apoptosis when exposed to the inhibitors, since these could influence the observed phenotype in a non-specific way. The English language needs at times to be improved.

Author Response

Point-by-point response to Reviewer 1

This MS by Vella et al describes the hyperactivation of the mTOR and STAT3 pathways in multiple lymphoid cell lineages as well as in bone marrow-derived mesenchymal stromal cells of Shwachman-Diamond syndrome patients, leading to overproduction of IL-6 which in turn could be involved in determining the high risk of myeloid neoplasia characterizing these patients. Additionally, they show that the mTOR inhibitor everolimus as well as mTOR or STAT3 silencing normalize STAT3 phosphorylation and IL-6 expression and secretion in several kinds of cells, suggesting that everolimus treatment could be of help in managing myeloid neoplasia development.

This is an interesting paper and the results are mostly sound.

R1.1. We thank the reviewer for his/her kind appreciation of our work

However, there are several problems as detailed below.

In several parts of the manuscript the authors state that mTOR can promote phosphorylation of STAT3 on both Y-705 and S-727 but the references cited do not show this at all (in particular, reference 28 does not even mention STAT3!). Moreover, as far as this reviewer can remember, while there are several papers showing the involvement of mTOR in S-727 phosphorylation, none mentions Y-P, as one would expect. Please use more carefully the literature and moderate these statements.

R1.2. We apologize for wrong citations. We fixed this issue adding a couple of references citing the involvement of mTOR in STAT3 S-727 phosphorylation. The Reviewer made a good point highlighting that this paper, together with our previous published paper (Bezzerri et al, SciRep 2016) are the only reports in which activation of mTOR is linked to STAT3 Y-705 phosphorylation. We discussed this more in the introduction (Page 3, Lines 96-99).

Fig 1: the authors state that mTOR phosphorylation is enhanced in all cell types analysed, while NK cells do not show any increase. Statistical analysis of data in Figs 1&2: why do the author use Mann-Whitney Rank Sum in Fig 1 and Student’s t test for paired data in Fig. 2? Please justify why not Mann-Whitney in both and unify the method. Moreover, the Kolmogorov-Smirnov test would perhaps be more suitable for both sets of experiments.

R.1.3. We agree with the reviewer that no significant difference of phospho-mTOR has been observed in NK. We corrected the sentence accordingly (Section 2.1. Page 5 Line 180). Statistical analysis of data in Figs 1&2 was performed accordingly to distribution analysis, which was previously performed by Shapiro Wilk test on each data group. Shapiro Wilk analysis revealed normal distribution for Fig.2, suggesting the use of parametric Student’s t test for pairing data. We thank the reviewer for pointing out this aspect. We clarified the data analysis and the use of Shapiro Wilk test both in methods (Page 15, Lines 682-685) and in the figure legends.

Finally, characters in both figures are too small and the figures themselves are too low resolution. Both legends to figure 1 and 2 lack a correct description of the graphics c and d.

R.1.4. We edited the figures including high-resolution images, accordingly. We apologize for the lack of a complete description of Figure 1 and 2, panels c and d. We fixed this issue.

Fig 3: why do the authors did not use the same samples for protein and mRNA detection of STAT3? Indeed, more samples analysed at the protein level would be desirable.

R.1.5. We thank the reviewer for the suggestion. Now mRNA and proteins have been quantified in both PBMC and LCL, expanding the number of patient samples tested. Briefly, we improved Figure 3 with: 1) new analysis of IL-6 mRNA in LCLs (new Fig. 3, panel a); 2) extension of STAT3 protein detection in several SDS patients (LCL) by WB analysis (new Fig. 3, panels b-d). Finally, we divided the original Figure 3 into two different figures (new Figs 3&4) to avoid cropped images.

Fig 4: please describe in the Methods how the bone marrow plasma was derived. Particularly since the authors emphasize the importance of the high IL6 secretion by BM-MSC making the point that the observed phenotype in patients could mainly be due to disrupted pathways in the stromal compartment of the bone marrow, it is essential to repeat the everolimus/STATTIC experiments of Fig. 5 and the siRNA experiments of Fig 6 in these cells as well.

R.1.6. We agree with the reviewer on the importance to extend the study of pharmacological inhibition on BM-MSC. Unfortunately, the bone marrow of SDS patients is often hypocellular, and we have not been able to collect more than 2 ml of BM from pediatric patients (also because of IRB restrictions). Thus, we cannot easily collect and analyze MSC. However, we did our best collecting four new BM-MSC samples from SDS patients and tested the effect of everolimus and stattic. New experimental results are shown in new Figure7 (panels e-f) and discussed within the main text. Surprisingly, although BM-MSC exhibit hyper-activation of mTOR (Bezzerri et al, 2016) and can express large amounts of IL-6, the treatment with both everolimus and stattic did not reduce IL-6 expression in these cells (discussed in Page 10, lines 397-399; Page 13, Lines 526-529). As suggested by reviewer 2, we found that sIL-6R levels are reduced in patients with SDS. Given the lack of membrane bound (mb)IL-6R expression in undifferentiated BM-MSC (Erices et al, 2002; Kondo et al, 2015), and deficiency of IL-6 “trans-signaling” in SDS, we hypothesize that undifferentiated BM-MSC are unaffected by the IL-6-mTOR-STAT3 feedforward loop of activation, which is instead displayed by hematopoietic cells. Importantly, the loss of sIL-6R signaling might also explain partially the impairment in ossification observed in SDS patients. This point has been discussed in the revised manuscript (Page 13, Lines 530-549).

All figure’s labels are too small and almost unreadable, please modify.

R.1.7. We increased the size of figure labels.

Finally, cells should be controlled for viability/apoptosis when exposed to the inhibitors, since these could influence the observed phenotype in a non-specific way. The English language needs at times to be improved.

R.1.8. We performed apoptosis assays on both control and SDS cells. As reported in new Supplementary Figure 3, everolimus did not lead to increased apoptosis nor necrosis. Stattic led instead to increased late apoptosis after 24 hours of treatment, although no effect was observed in early apoptosis and necrosis rates. Please note that we used a concentration of stattic (7.5 mM) which has been reported not to affect cell viability in human leukocytes (Severin et al, 2019). However, this result could be expected, because STAT3 knock out is lethal both in cell lines and animals, and JAK/STAT inhibitors are currently used as chemotherapeutic agents in cancer and leukemia. We discussed this finding in conclusions (Page 13, Lines 517-529). English language has been internally reviewed by English-native speaker.

Reviewer 2 Report

The article studies lymphocyte populations in Shwachman-Diamond-Syndrome, an inherited bone marrow failure syndrome. The authors analyzed patient materials and patient cells for the activation of the previously noted mTOR-STAT3-IL-6 axis, which they show to be hyperactivated in the lymphoid compartment resulting in massive IL-6 production. Everolimus mediated inhibition of mTOR, knock-down of mTOR and knock-down of STAT3 alleviated this hyperactivation. The authors speculate that inhibition of mTOR by everolimus might be useful for the treatment of Shwachman-Diamond-Syndrome or similar malignancies.

This is a very interesting study with relevant results. There are, however, some questions the authors should address.

Major points:

There is a discrepancy in the IL-6 levels reported in the study. In line 163, the authors state that IL-6 levels in in primary SDS BM-MSCs are around 8 ng/ml per 105 In Fig. 4, however, it is stated that in the supernatant of 106 cells after 48 h, around 8000 pg/ml IL-6 were detected. In line 243 in the discussion section, again 8 ng/ml IL-6 per 105 cells is stated. This needs to be reconciled. It has been shown that not only IL-6 but also the soluble IL-6R is important to judge the biologic activity of IL-6. For instance, T-cells quantitatively shed the IL-6R upon stimulation and are thereafter only responsive to the combination of IL-6 and soluble IL-6R. Therefore, it would be informative if soluble IL-6R levels were measured and the most relevant role of the soluble IL-6R would be explained.

Minor points:

The lettering in the FACS panels of Fig. 1,2 is too small and very hard to read. This reviewer does not understand the expression: '…almost the totality of T cell subpopulations…' (line 229). This reviewer does not understand the sentence: '…that resulted significantly higher than age-matched healthy controls…' (line 249). Some references are incomplete.

Author Response

Point-by-point response to Reviewer 2

The article studies lymphocyte populations in Shwachman-Diamond-Syndrome, an inherited bone marrow failure syndrome. The authors analyzed patient materials and patient cells for the activation of the previously noted mTOR-STAT3-IL-6 axis, which they show to be hyperactivated in the lymphoid compartment resulting in massive IL-6 production. Everolimus mediated inhibition of mTOR, knock-down of mTOR and knock-down of STAT3 alleviated this hyperactivation. The authors speculate that inhibition of mTOR by everolimus might be useful for the treatment of Shwachman-Diamond-Syndrome or similar malignancies.

This is a very interesting study with relevant results.

R.2.1. We thank the reviewer for his/her kind appreciation of our work.

There are, however, some questions the authors should address.

Major points:

There is a discrepancy in the IL-6 levels reported in the study. In line 163, the authors state that IL-6 levels in primary SDS BM-MSCs are around 8 ng/ml per 105 In Fig. 4, however, it is stated that in the supernatant of 106 cells after 48 h, around 8000 pg/ml IL-6 were detected.In line 243 in the discussion section, again 8 ng/ml IL-6 per 105 cells is stated. This needs to be reconciled.

R.2.2. We apologize for this inconsistency. MSC were seeded at a density of 2x105 cells. We corrected this discrepancy within the text and in the figure legends.

It has been shown that not only IL-6 but also the soluble IL-6R is important to judge the biologic activity of IL-6. For instance, T-cells quantitatively shed the IL-6R upon stimulation and are thereafter only responsive to the combination of IL-6 and soluble IL-6R. Therefore, it would be informative if soluble IL-6R levels were measured and the most relevant role of the soluble IL-6R would be explained.

R.2.3. We thank the reviewer for the useful suggestion. We indeed quantified soluble IL-6R in peripheral blood and bone marrow plasma samples obtained from an enlarged cohort of SDS patients. Interestingly, we found that sIL-6R is reduced in plasma samples of SDS patients compared to age-matched healthy controls. These data support a reduced IL-6 “trans-signaling” in SDS that might at least partially justify the lack of response to everolimus and stattic treatment by undifferentiated BM-MSC (which do not express mbIL-6R as well) and the delayed ossification which is reported in SDS. We extensively discussed these findings in discussion (Page 13, Lines 530-549).

Minor points:

The lettering in the FACS panels of Fig. 1,2 is too small and very hard to read. This reviewer does not understand the expression: '…almost the totality of T cell subpopulations…' (line 229). This reviewer does not understand the sentence: '…that resulted significantly higher than age-matched healthy controls…' (line 249). Some references are incomplete.

R.2.4. We have rephrased the text and adjusted the references.
